# The Complete Chloroplast Genome Sequence of the *Speirantha gardenii*: Comparative and Adaptive Evolutionary Analysis

**Gurusamy Raman *** and **SeonJoo Park ***

Department of Life Sciences, Yeungnam University, Gyeongsan 38541, Gyeongsan-buk, Korea

* Correspondence: bioramg@yu.ac.kr (G.R.); sjpark01@yu.ac.kr (S.P.); Tel.: +82-53-810-3865 (G.R.); +82-53-810-2377 (S.P.); Fax: +82-53-810-4618 (S.P.)

**Abstract:** The plant "False Lily of the Valley", *Speirantha gardenii* is restricted to south-east China and considered as an endemic plant. Due to its limited availability, this plant was less studied. Hence, this study is focused on its molecular studies, where we have sequenced the complete chloroplast genome of *S. gardenii* and this is the first report on the chloroplast genome sequence of *Speirantha*. The complete *S. gardenii* chloroplast genome is of 156,869 bp in length with 37.6% GC, which included a pair of inverted repeats (IRs) each of 26,437 bp that separated a large single-copy (LSC) region of 85,368 bp and a small single-copy (SSC) region of 18,627 bp. The chloroplast genome comprises 81 protein-coding genes, 30 tRNA and four rRNA unique genes. Furthermore, a total of 699 repeats and 805 simple-sequence repeats (SSRs) markers are identified in the genome. Additionally, $K_A/K_S$ nucleotide substitution analysis showed that seven protein-coding genes have highly diverged and identified nine amino acid sites under potentially positive selection in these genes. Phylogenetic analyses suggest that *S. gardenii* species has a closer genetic relationship to the *Reineckea*, *Rohdea* and *Convallaria* genera. The present study will provide insights into developing a lineage-specific marker for genetic diversity and gene evolution studies in the Nolinoideae taxa.

**Keywords:** *Speirantha gardenii*; chloroplast genome; positive selection; adaptive evolution; substitution; Nolinoideae

## 1. Introduction

The plant chloroplast plays a pivotal role in photosynthesis and other biological metabolic processes that mediate the adaptation of the plant to the surrounding environment [1]. The highly conserved angiosperm plants encode a circular chloroplast genome with a quadripartite structure, consists of a large single-copy (LSC) region, and small single-copy (SSC) region which is separated by a duplicate inverted repeat (IRa and IRb) regions and differences in genome size and composition are taxonomically informative [1–5]. Although the amount of variation is not very significant across flowering plants, the chloroplast genome size varies between species that ranges from 107 kb (*Cathaya argyrophylla*) to 280 kb (*Pelargonium*) [6,7]. Normally the chloroplast genome encodes 120 to 130 genes, involved in photosynthesis, transcription and translation process [6]. Though the angiosperm chloroplast genomes are highly conserved, several mutational events, such as structural rearrangement, insertions and deletions (InDels), inversions, translocations, and copy number variations (CNVs) occur within the chloroplast genomes. This polymorphism in the chloroplast genome provides an understanding of population genetics, phylogenetic and evolutionary studies, species barcoding and endangered species conservation and enhancement of breeding of the plants.

Flowering plants are the largest clade among the land plants, consisting of more than 250,000 species [8]. Among these, Nolinoideae is a subfamily, with more than 100 species, of the

Asparagaceae family belonging to the monocot flowering plants. In the past decade, a few species of Nolinoideae have been characterized at the molecular level and phylogenetic implications with other species were identified [9–15]. The genus *Speirantha* belongs to the subfamily Nolinoideae and consists of only one species, *S. gardenii* and its common name is "False Lily of the Valley". The distribution of this species is restricted to south-east China and is considered an endemic plant [16]. It is a delightful and intriguing small-scale evergreen perennial plant with panicles of delicate starry white flowers during early spring and the foliage is glossy, pale green and elliptic or elliptic-oblanceolate in shape. Due to its rare availability, extensive molecular studies have not been carried out for this species. So, in the present study, we report the first complete chloroplast genome sequence of *S. gardenii* and analyzed repeat regions and simple-sequence repeats (SSRs) markers in the genome. Furthermore, we compared the *S. gardenii* chloroplast genome with its closely related species. Additionally, highly variable regions and seven protein-coding genes that are discovered in their genome to be under positive selection, could be employed to create potential markers for phylogenetic studies or candidates for DNA barcoding in future studies.

## 2. Results

### 2.1. General Features of the Speirantha Gardenii Chloroplast Genome

The overall length of the *S. gardenii* chloroplast genome is 156,869 bp, exhibiting the circular quadripartite structure characteristic of major angiosperm plants. The chloroplast genome consists of a pair of the inverted repeat (IR) regions (26,437 bp) separated by a large single-copy (LSC) region of 85,368 bp and a small single-copy (SSC) region of 18,627 bp (Figure 1). When calculating duplicated genes in the IR region only one time, the chloroplast genome contains 115 genes, including 81 protein-coding genes, 30 tRNA genes and four rRNA genes (Table 1). All four rRNAs, nine protein-coding genes and eight tRNA genes are duplicated in the IR regions, making the total number of 136 genes. Seventeen genes contain introns, including five tRNA and ten protein-coding genes with a single intron and *clpP* and *ycf3* with two introns (Supplementary Materials Table S1). Overall, the order and contents of the gene of the *S. gardenii* are identical with other species of Nolinoideae except the length of the *infA* gene and pseudogene *infA* in the *C. keiskei*, *Liriope spicata* and *Nolina atopocarpa* (Supplementary Materials Table S2). The GC content of the *S. gardenii* chloroplast genome is 37.6% (Table 1), like *R. carnea*, whereas GC content is low in the species of *R. chinensis* (37.2%) and high in the *C. keiskei* (37.9%).

**Table 1.** The characteristic feature of the *Speirantha gardenii* chloroplast genome.

| Sequence Region | | S. gardenii |
|---|---|---|
| Total chloroplast genome size (bp) | | 156,869 |
| LSC length (bp) | | 85,368 |
| SSC length (bp) | | 18,627 |
| IR length (bp) | | 26,437 |
| Total number of genes | | 136 |
| Protein-coding genes | | 90 |
| tRNA genes | | 38 |
| rRNA genes | | 8 |
| Genes duplicated by IR | | 21 |
| Genes with introns | | 17 |
| GC content | Total (%) | 37.6 |
| | LSC (%) | 35.6 |
| | SSC (%) | 31.5 |
| | IR (%) | 43.0 |
| | CDS (%) | 38.1 |
| | rRNA (%) | 55.3 |
| | tRNA (%) | 53.2 |
| | All genes (%) | 39.8 |
| Protein-coding genes (%bp) | | 50.13 |
| All genes (%bp) | | 71.42 |
| Non-coding regions (%) | | 28.58 |

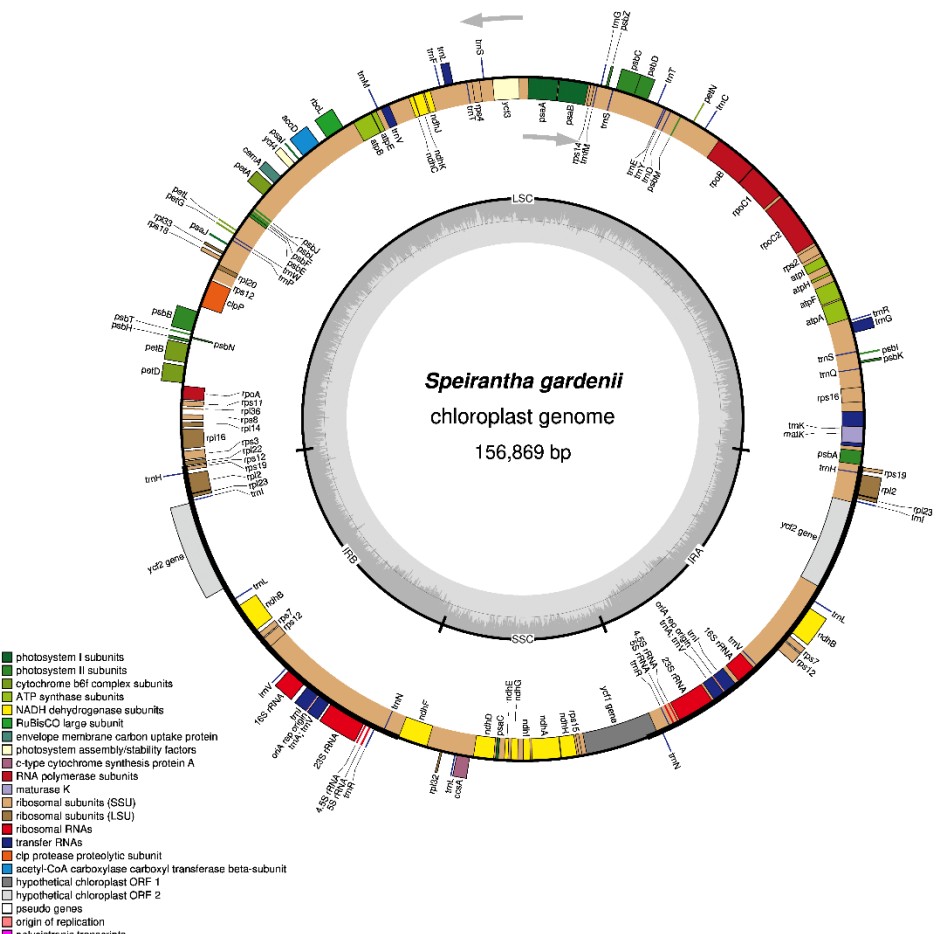

**Figure 1.** Gene map of *Speirantha gardenii*. Genes lying outside the outer circle are transcribed in a counter-clockwise direction, and genes inside this circle are transcribed in a clockwise direction. The colored bars indicate known protein-coding genes, transfer RNA genes, and ribosomal RNA genes. The dashed, dark grey area in the inner circle denotes GC content, and the light grey area indicates genome AT content. LSC, large single-copy; SSC, small single-copy; IR, inverted repeat.

*2.2. Comparative Analysis of the IR Contraction and Expansion in the Species of Nolinoideae*

The LSC-IR and SSC-IR borders of the *S. gardenii* chloroplast genome are compared with three other closely related species (*R. carnea*, *R. chinensis* and *C. keiskei*) of the Nolinoideae subfamily (Figure 2). Two intact copies of the *rps19* gene are present in the IR regions of all chloroplast genomes, whereas, in the IRbSSC border, the pseudogene *yfc1* and *ndhF* gene crosses the IRb/SSC border region and overlaps with 1–34 bp region in the borders. Similarly, the intact *ycf1* gene in all the chloroplast genomes except *R. chinensis* crosses SSC/IRa region with an 827–913 bp length fragment of *ycf1* located in the IRA region. In contrast, the functional and pseudogene of *ycf1* of the *R. chinensis* have dispersed in the SSC region and 308 bp away from the IRb/SSC and SSC/IRa border. Due to this ψ*ycf1* gene shift in the IRb/SSC border of *R. chinensis* chloroplast genome, the gene *ndhF* is present in the SSC region and the *trnN* gene is present in the SSC/IRa region. The *psbA* gene sequences are found in LSC regions in all the chloroplast genomes. This gene is ~81–82 bp away from the IRa/LSC border of *S. gardenii*, *R. carnea*, and *C. keiskei* chloroplast genomes but 331 bp away for *R. chinensis*.

The sequence variation in chloroplast genomes of four Nolinoideae subfamily chloroplast genomes is plotted using the mVISTA program, where minor divergence is identified between *S. gardenii* and *R. carnea* (Figure 3). When *S. gardenii* is compared with two other chloroplast genomes namely

*R. chinensis* and *C. keiskei*, it is highly diverged due to the presence of sequence variation in both protein-coding and intergenic regions of these two species.

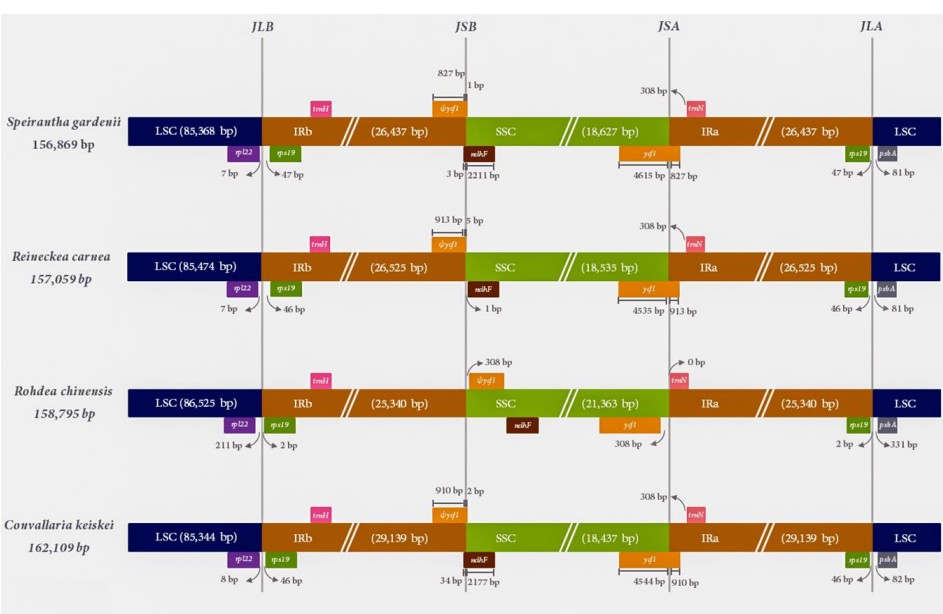

**Figure 2.** Comparison of the large single-copy (LSC), small single-copy (SSC) and inverted repeat (IR) border regions of four Nolinoideae chloroplast genomes. ψ indicates a pseudogene. The figure is not drawn to scale.

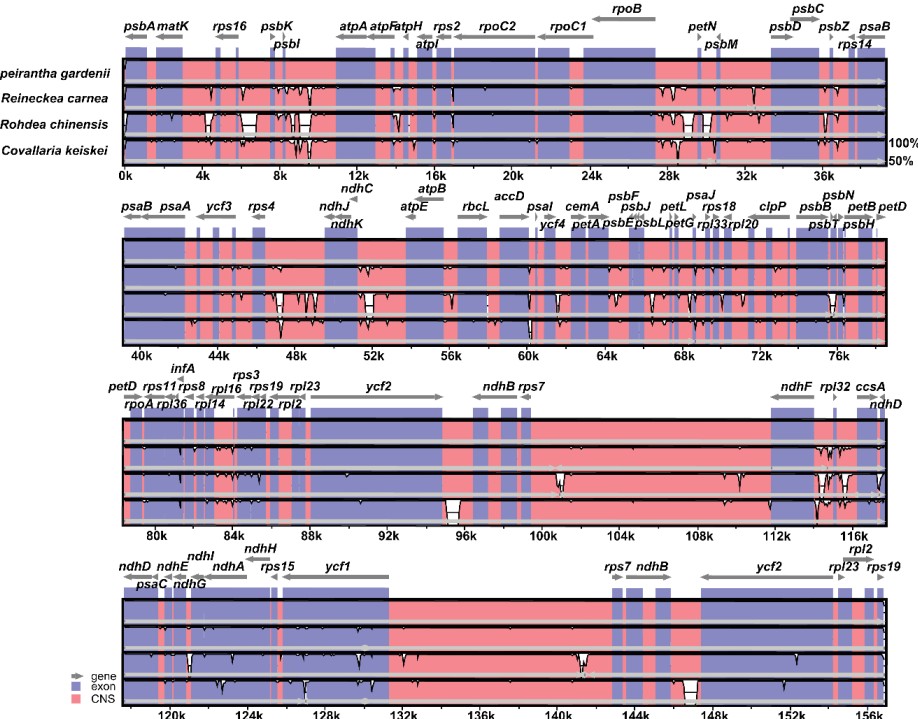

**Figure 3.** Sequence alignment of four Nolinoideae chloroplast genomes performed using the mVISTA program with *Speirantha gardenii* as a reference. The top grey arrow shows genes in order (transcriptional direction) and the position of each gene. A 70% cut-off was used for the plots. The y-axis indicates a percent identity of between 50% and 100%, and the red and blue areas indicate intergenic and genic regions, respectively.

## 2.3. Synonymous (K_S) and Nonsynonymous (K_A) Substitution Rate Analysis

Synonymous and nonsynonymous substitution rates are analyzed for 79 protein-coding genes of *S. gardenii*, *R. carnea*, *R. chinensis* and *C. keiskei* chloroplast genomes (Figure 4). The $K_A/K_S$ ratio of most of the genes are less than 1, except *ccsA* (1.13–1.43), *infA* (1.19), *ndhF* (1.36–2.15), *rpl20* (1.59), *rps2* (1.48), *rps3* (1.18–1.76) and *ycf1* (1.13–1.23) protein-coding genes.

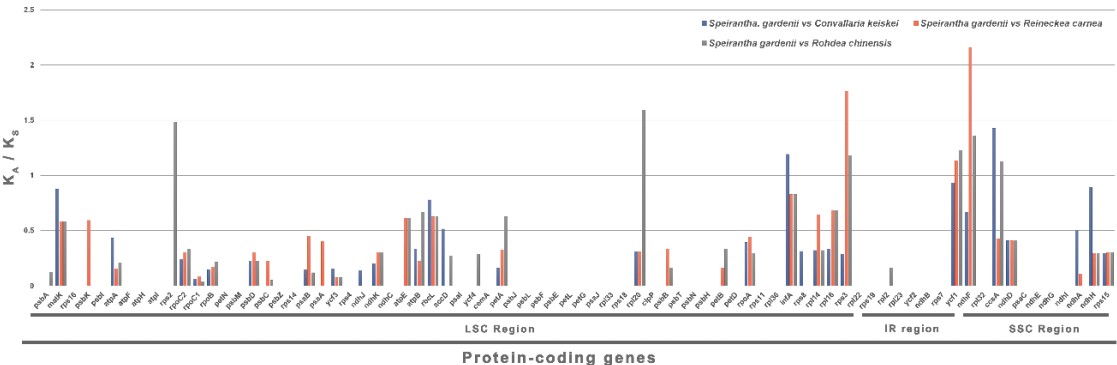

**Figure 4.** The ratio of non-synonymous ($K_A$) to synonymous ($K_S$) substitutions of 79 protein-coding genes of four species of Nolinoideae.

## 2.4. Selective Pressure Events

The positive selection of seven protein-coding genes (*ccsA*, *infA*, *ndhF*, *rpl20*, *rps2*, *rps3* and *ycf1*), genes of four closely related species (*S. gardenii*, *R. carnea*, *R. chinensis* and *C. keiskei*) and publicly available Nolinoideae chloroplast genome species were analyzed separately. The $\omega_2$ values of seven genes of four Nolinoideae species are ranging from 1 to 999 in the M2a model. So further, we compared these seven genes across Nolinoideae species to understand the selective pressure events. Due to the presence of pseudogenization of the *infA* and *ycf1* gene in some Nolinoideae species, we analyzed the remaining five genes and identified the $\omega_2$ values ranging from 1 to 240.876. Furthermore, Bayes empirical Bayes (BEB) analysis is used to analyze the location of consistent selective sites in the seven protein-coding genes of four Nolinoideae species using the M7 vs. M8 model. The analysis revealed that five sites under potentially positive selection in the three protein-coding genes (*infA*-3; *ndhF*-1 and *rps3*-1) with posterior probabilities more than 0.95. Furthermore, the two sites (*rps2*-1 and *ycf1*-1) with greater than 0.99 (Supplementary Table S3) and the 2ΔLnL value is ranging from 0 to 26.084 (Table 2). Additionally, five genes (*ccsA*, *ndhF*, *rpl20*, *rps2* and *rps3*) of all Nolinoideae species were analyzed and predicted that seven sites (*ccsA*-3 and *ndhF*-4) greater than 0.95 and two sites (*ndhF*-1 and *rps2*-1) with >0.99 (Supplementary Table S4) and the 2ΔLnL value is ranging from 0 to 108.245 (Table 3). In both analyses revealed that *rpl20* does not encode any positively selected sites in their gene and the 2ΔLnL and the *p*-value of LRT are 0 to 0.40 and 0.895–1.0, respectively.

**Table 2.** Comparison of likelihood ratio test (LRT) statistics of positive selection models against their null models (2ΔLnL) and positive selective amino acid loci for four Nolinoideae species.

| Protein-Coding Genes | Comparison between Models | 2ΔLnL | d.f. | p-Value | Positive Sites (M7 vs. M8) |
|---|---|---|---|---|---|
| *ccsA* | M0 vs. M3 | 0 | 4 | 1 | 81 N 0.642, 125 S 0.642, 223 S 0.642, 280 S 0.642, 292 I 0.636 |
| | M1 vs. M2A | 0 | 2 | 1 | |
| | M7 vs. M8 | 0.000002 | 2 | 0.999999 | |
| | M8a vs. M8 | 0.001524 | 1 | 0.968859739 | |
| *infA* | M0 vs. M3 | 15.845366 | 4 | 0.003233836 | 32 L 0.759, 56 I 0.881, 57 G 0.966 *, 58 M 0.965 *, 59 Q 0.737, 60 L 0.961 * |
| | M1 vs. M2A | 11.350176 | 2 | 0.003430367 | |
| | M7 vs. M8 | 11.352554 | 2 | 0.003426291 | |
| | M8a vs. M8 | 11.350166 | 1 | 0.000754412 | |

**Table 2.** *Cont.*

| Protein-Coding Genes | Comparison between Models | 2ΔLnL | d.f. | p-Value | Positive Sites (M7 vs. M8) |
|---|---|---|---|---|---|
| *ndhF* | M0 vs. M3 | 12.435874 | 4 | 0.014387898 | 67 F 0.543, 281 F 0.568, 472 Y 0.581, 486 E 0.606, 505 N 0.571, 560 H 0.583, 574 L 0.617, 596 L 0.665, 614 S 0.616, 636 G 0.632, 675 L 0.580, 680 Q 0.628, 728 F 0.568, 732 L 0.963 *, 734 F 0.931 |
| | M1 vs. M2A | 9.771726 | 2 | 0.007552603 | |
| | M7 vs. M8 | 8.535662 | 2 | 0.014012143 | |
| | M8a vs. M8 | 8.529956 | 1 | 0.003493481 | |
| *rpl20* | M0 vs. M3 | 0 | 4 | 1 | 43 L 0.620, 80 R 0.612, 116 M 0.612, 117 K 0.603 |
| | M1 vs. M2A | 0 | 2 | 1 | |
| | M7 vs. M8 | 0.00013 | 2 | 0.999935002 | |
| | M8a vs. M8 | 0.00004 | 1 | 0.994953769 | |
| *rps2* | M0 vs. M3 | 24.069072 | 4 | 0.000077368 | 48 T 0.838, 59 D 0.839, 91 A 0.838, 160 E 0.840, 199 L 0.850, 237—0.998 ** |
| | M1 vs. M2A | 23.735088 | 2 | 0.000007014 | |
| | M7 vs. M8 | 15.316822 | 2 | 0.000472057 | |
| | M8a vs. M8 | 13.715290 | 1 | 0.000212716 | |
| *rps3* | M0 vs. M3 | 17.514528 | 4 | 0.001534958 | 28 N 0.937, 30 S 0.599, 68 Q 0.584, 105 F 0.608, 106 H 0.580, 221—0.988 * |
| | M1 vs. M2A | 12.631522 | 2 | 0.00180759 | |
| | M7 vs. M8 | 14.593672 | 2 | 0.00067768 | |
| | M8a vs. M8 | 13.505938 | 1 | 0.00023781 | |
| *ycf1* | M0 vs. M3 | 26.084252 | 4 | 0.000030431 | 267 Y 0.939, 271 Y 0.517, 290 D 0.520, 296 Y 0.995 **, 407 L 0.523, 438 R 0.536, 447 S 0.503, 491 T 0.545, 507 Q 0.503, 558 K 0.920, 601 I 0.539, 804 L 0.523, 864 I 0.503, 921 F 0.535, 953 R 0.936, 1068 S 0.558, 1082 Q 0.927, 1096 S 0.516, 1108 R 0.935, 1162 R 0.510, 1255 L 0.559, 1340 L 0.926, 1370 Q 0.524, 1373 Q 0.503, 1375 F 0.515, 1484 Q 0.509, 1499 I 0.523, 1526 F 0.541, 1543 L 0.505, 1639 Q 0.530, 1671 H 0.925, 1672 F 0.515 |
| | M1 vs. M2A | 19.806836 | 2 | 0.000050003 | |
| | M7 vs. M8 | 19.228316 | 2 | 0.000066777 | |
| | M8a vs. M8 | 18.528045 | 1 | 0.000016742 | |

Positively selected sites (* *p* > 95%; ** *p* > 99%).

**Table 3.** Comparison of likelihood ratio test (LRT) statistics of positive selection models against their null models (2ΔLnL) and positive selective amino acid loci for across all Nolinoideae species.

| Protein-Coding Genes | Comparison between Models | 2ΔLnL | d.f. | p-Value | Positive Sites (M7 vs. M8) |
|---|---|---|---|---|---|
| *ccsA* | M0 vs. M3 | 25.070924 | 4 | 0.000048685 | 170 D 0.989 *, 175 R 0.969 *, 178 F 0.764, 184 F 0.978 *, 186 D 0.871, 206 R 0.868, 278 S 0.582 |
| | M1 vs. M2A | 13.324234 | 2 | 0.001278437 | |
| | M7 vs. M8 | 13.687792 | 2 | 0.001065942 | |
| | M8a vs. M8 | 13.319022 | 1 | 0.000262727 | |
| *ndhF* | M0 vs. M3 | 74.589864 | 4 | 0 | 77 V 0.814, 463 K 0.804, 486 A 0.610, 513 G 0.702, 514 R 0.638, 523 H 0.828, 531 T 0.942, 546 V 0.979 *, 560 N 0.998 **, 584 P 0.726, 586 F 0.947, 588 G 0.550, 590 P 0.961 *, 596 L 0.698, 636 G 0.975 *, 638 P 0.850, 675 L 0.637, 680 Q 0.871, 728 F 0.830, 729 F 0.720, 732 L 0.973 *, 733 F 0.627 |
| | M1 vs. M2A | 17.810044 | 2 | 0.000135706 | |
| | M7 vs. M8 | 18.041236 | 2 | 0.000120891 | |
| | M8a vs. M8 | 17.827478 | 1 | 0.000024187 | |
| *rpl20* | M0 vs. M3 | 0.407316 | 4 | 0.98187355 | 76 Y 0.698 |
| | M1 vs. M2A | 0.016874 | 2 | 0.991598492 | |
| | M7 vs. M8 | 0.067732 | 2 | 0.966701034 | |
| | M8a vs. M8 | 0.01735 | 1 | 0.895206241 | |
| *rps2* | M0 vs. M3 | 108.22369 | 4 | 0 | 32 A 0.524, 131 N 0.504, 199 L 0.612, 237—1.000 ** |
| | M1 vs. M2A | 108.22376 | 2 | 0 | |
| | M7 vs. M8 | 108.237424 | 2 | 0 | |
| | M8a vs. M8 | 108.245618 | 1 | 0 | |
| *rps3* | M0 vs. M3 | 2.187614 | 4 | 0.70129752 | 30 S 0.571, 86 E 0.568 |
| | M1 vs. M2A | 0 | 2 | 1 | |
| | M7 vs. M8 | 0.000004 | 2 | 0.999998 | |
| | M8a vs. M8 | 0.025852 | 1 | 0.87226229 | |

Positively selected sites (* *p* > 95%; ** *p* > 99%).

### 2.5. Analysis of infA Gene

The translation factor IF-1 (*infA*) gene of *S. gardenii* is compared with the other three Nolinoideae species. The comparative analysis showed that 26 bp deleted at the 3' end of the *infA* gene followed by 79 bp deletion in the intergenic region between *infA* and *rpl36* gene in the chloroplast genome of *S. gardenii*. Due to this frameshift mutation, the 3' end of the *infA* is extended and 26 bp overlaps with the *rpl36* gene, thus leads to the length of the *infA* gene increased to 282 bp (Figure 5).

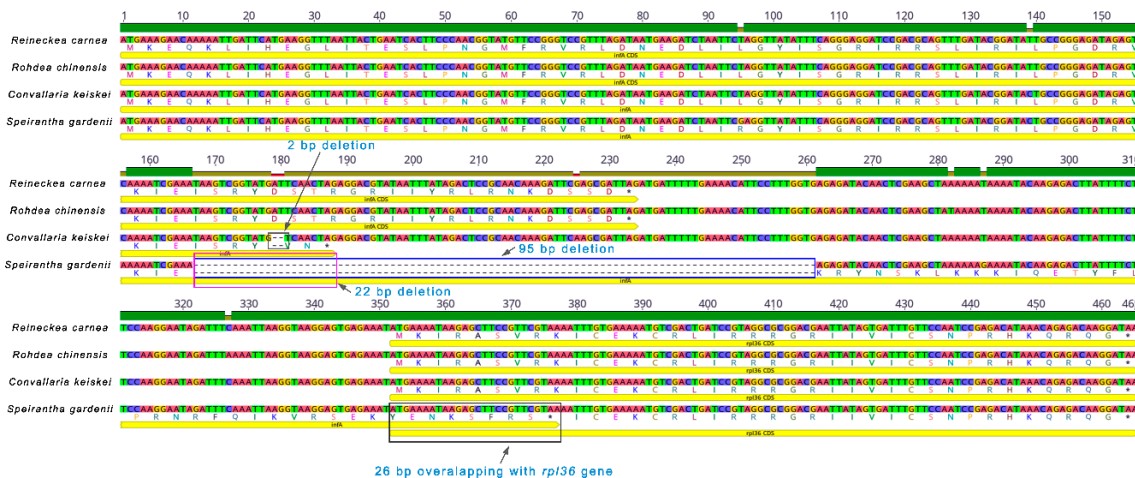

**Figure 5.** Comparison of *infA* and *rpl36* gene and their intergenic regions of four Nolinoideae species.

### 2.6. Repeat Sequence and Simple Sequence Repeat Analysis

REPuter program is used to determine the presence of repeat sequences in the *S. gardenii* chloroplast genome. The analysis showed that a total of 699 repeats, with motif length from 30 to 115 bp are present in its genome. The repeats sequences included 264 direct, 251 reverse, 228 complementary and 256 palindromic repeats (Figure 6a). Of these repeats, 30–39 bp long repeats predominantly occupy in the chloroplast genome and account for 98.1% (680 repeats) (data not shown). The remaining 19 repeats are distributed in the range of 40 to 115 bp length (Figure 6b).

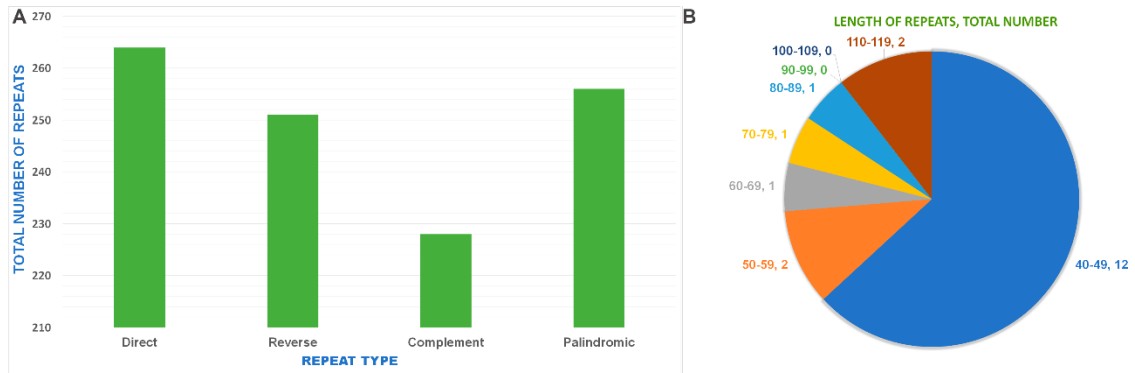

**Figure 6.** The distribution of different repeat types in the *Speirantha gardenii* chloroplast genome. (**A**) The number of different types of repeats. F—forward repeats; R—reverse repeats; P—palindromic repeats; C—complement repeats (**B**). The length and the total number of repeat sequences present in the chloroplast genome.

A total of 805 simple sequence repeats (SSRs) were identified in the *S. gardenii* chloroplast genome. Of these, 268 (33.29%) are mono-nucleotide repeats, 37 (5.6%) di-nucleotide repeats, 63 (7.8%) tri-nucleotide repeats, 84 (10.4%) tetra-nucleotide repeats, 122 (15.15%) penta-nucleotide repeats, 140 (17.39%) hexa-nucleotide repeats, 37 (4.6%) 7-nucleotide repeats, 16 (1.99%) 8-nucleotide repeats,

18 (2.24%) 9-nucleotide repeats and 4, 5, 4, 1, 2, 2 and 2 are 10-, 11-, 12-, 14-, 16-, 17- and 22- nucleotide repeats, respectively (Figure 7a). Of the 805 SSRs, 72.17% (581), 11.55% (93) and 16.28% (131) SSRs are present in the LSC, IR, and SSC regions, respectively (Figure 7b). Additionally, the distribution of SSR in the protein-coding, intron and intergenic regions (IGS) were analyzed, and found that 517 (64.22%), 208 (25.84%), and 80 (9.94%) SSRs are located in IGS, protein-coding, and intron regions, respectively (Figure 7c).

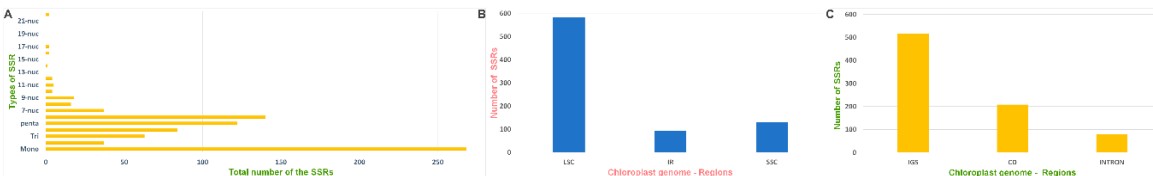

**Figure 7.** The presence of simple sequence repeats (SSRs) in the *Speirantha gardenii* chloroplast genome. (**A**) Distribution of different types of SSRs; (**B**) Presence of SSRs in the LSC, SSC, and IR regions; (**C**) Presence of SSRs in intergenic spacers, protein-coding regions, and intron regions.

*2.7. Phylogenetic Analysis*

To analyze the phylogenetic position of *S. gardenii* with other Nolinoideae species, 75 protein-coding genes of 19 Nolinoideae chloroplast genomes are aligned. ML phylogenetic tree analysis revealed that Nolinoideae species formed a monophyletic group (Figure 8). *S. gardenii* clustered with *R. carnea*, *R. chinensis* and *C. keiskei* with strong bootstrap value and showed that *S. gardenii* is sister to both *R. carnea*, *R. chinensis* and *C. keiskei*.

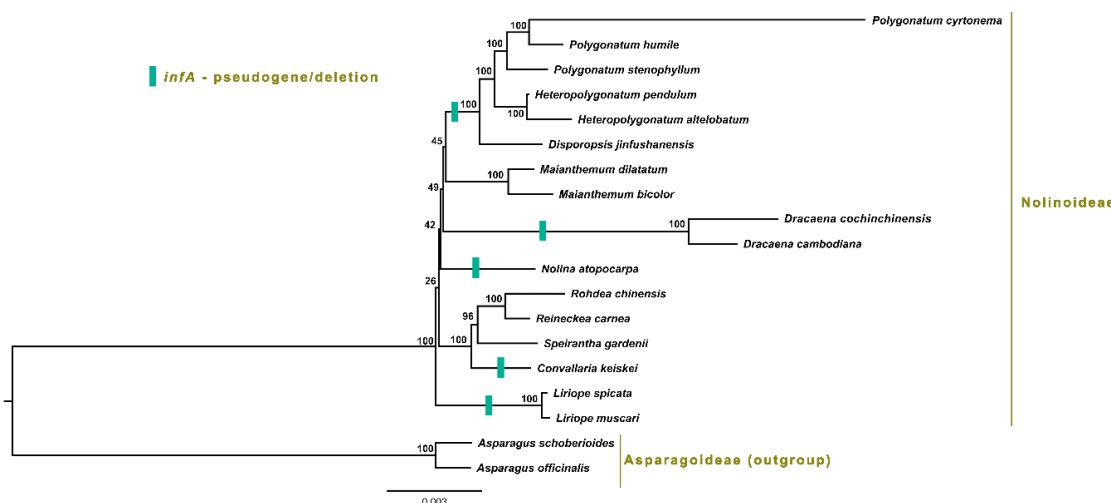

**Figure 8.** Molecular phylogenetic tree based on 75 protein-coding genes of 19 Asparagaceae chloroplast genomes. *Asparagus* set as the outgroup. The tree was constructed by maximum likelihood (ML) analysis of the conserved regions using the RAxML program and the general time-reversible invariant-sites (GTRI) nucleotide substitution model. The stability of each tree node was tested by bootstrap analysis with 1000 replicates. Bootstrap values are indicated on the branches, and the branch length reflects the estimated number of substitutions per 1000 sites. The Puerto Rico rectangular box indicates that the *infA* gene is either a pseudogene or deleted in their respective chloroplast genome.

## 3. Discussion

The species *Speirantha gardenii* is a monocot plant of the Nolinoideae subfamily of the family Asparagaceae. Until recently, only a few complete chloroplast genome sequences for this Nolinoideae subfamily have been deposited in GenBank, with the very first being that of *Polygonatum cyrtonema* in

2015. Owing to the development of high-throughput sequencing technologies, additional Nolinoideae chloroplast genomes were sequenced [9–15], but the genus *Speirantha* has to date remained unexplored. Therefore, in the present study, we sequenced the *S. gardenii* chloroplast genome and compared it with its closed related other Nolinoideae members. The length of the complete chloroplast genome sequence of *S. gardenii* is 156,869 bp and contains 137 individual genes, which is in the range of other Nolinoideae and angiosperms. The GC content of *S. gardenii* is 36.7%, which is similar with *R. carnea*, but differs with other closely related species, namely, *R. chinensis* (37.2%) and *C. keiskei* (37.9%) suggesting that the distribution of the GC contents in the Nolinoideae chloroplast genomes are inconsistent and this difference is due to the presence of high-level GC nucleotide percentages in the four rRNA genes in IR regions (Table 1). Likewise, similar results have been identified in other angiosperm chloroplast genomes [17–19].

Though the gene order and gene content are similar to other Nolinoideae species, the length of the chloroplast genome differs in the Nolinoideae species. LSC region of the species of Nolinoideae subfamily is generally similar but differs in SSC and IR regions in some species. The SSC region of the *R. chinensis* (21,363 bp) is expanded due to the shift of the *ycf1* gene from the IR/SSC border to the SSC region. To support this event, plenty of insertion and deletion process is observed in their chloroplast genome. Interestingly, the IR region of the *C. keiskei* is expanded due to the integration of 3.3 kb mitochondrial region in the chloroplast genome and this event is restricted to the *Convallaria* genus [12].

Furthermore, the high sequence variation is identified in the chloroplast genomes of Nolinoideae species. When *S. gardenii* is compared with other closely related species, the minor divergence is identified in the intergenic regions of the *R. carnea* chloroplast genome. Besides, high sequence variation is identified in the intergenic regions of *C. keiskei* and intergenic and protein-coding regions of *R. chinensis*. This sequence variation is due to the insertion of the mitochondrial region in the chloroplast genome of *C. keiskei* and indel observed in the intergenic and protein-coding regions of *R. chinensis* genome. Owing to the indel events in the Nolinoideae species, the substitution rate impacts in some protein-coding genes. The rate of synonymous substitutions ($K_S$) accumulates nearly neutral evolution, whereas the rate of nonsynonymous substitutions ($K_A$) are subjected to selective pressures of varying degree and positive or negative direction [20]. The ratio of $K_A/K_S$ ($\omega$) value below 1 indicates that the corresponding genes experiencing relaxed or purifying selection while $\omega = 1$ and $\omega > 1$ indicate neutral and positive selection, respectively [20,21]. The $K_A/K_S$ ratio of the protein-coding genes, such as *ccsA*, *infA*, *ndhF*, *rpl20*, *rps2*, *rps3* and *ycf1*, are more than 1, which indicates that these genes are under positive selection (Figure 4). This deviation from unity is due to the presence of indel, premature stop codon and amino-acid substitution events in the protein-coding genes, such as the 6 bp deletion and three amino-acid change in the *ccsA* and *infA* of *C. keiskei*; 6 and 5 amino acid change by non-synonymous substitution in the *ndhF* gene of *R. carnea* and *R. chinensis*; 8 bp insertion and 2 amino-acid substitution in the *rpl20* and *rps2* of *R. chinensis*; 3 amino acid change in the *rps3* of *R. carnea*; 18 amino-acid change and 161 bp insertion in the *ycf1* gene of *R. carnea* and *R. chinensis*. Though we have identified high $K_A/K_S$ nucleotide substitution ratio in these protein-coding genes, the overall nucleotide identity is >99.2%, except for *infA* (85%). So, we compared the *infA* gene in the Nolinoideae subfamily. Usually, the length of the *infA* gene is 234 bp. Most of the species of the Nolinoideae subfamily do not contain an *infA* gene or contain it as a pseudogene except *Maianthemum dilatatum*, *M. bicolor*, *R. carnea* and *R. chinensis* [9,10,13]. Meanwhile, the deletion of two bp at the 180th bp position in the *infA* gene of *C. keiskei*, causes frameshift mutation to a premature stop codon and thus leads to the formation of *infA* pseudogene [12]. In contrast, a total of 95 bp deletion occurred at the 3′ end of the *infA* gene (26 bp) and the intergenic region between the *infA* and *rpl36* gene (79 bp). Due to the indel process, a frameshift mutation occurred and the 3′ end of the *infA* is extended and 26 bp overlaps with the *rpl36* gene, which thus led to the length of the *infA* gene increased to 282 bp (Figure 5). Furthermore, a similar type of elongated *infA* gene is observed in many monocot plants such as *Zea mays*, *Oryza sativa*, *Hordeum vulgare* etc. [22–24]. Besides, previous studies also revealed that most of the angiosperms have lost their *infA* gene independently in their chloroplast genome [25].

Based on ω analysis of 79 protein-coding genes of *S. gardenii*, *R. carnea*, *R. chinensis* and *C. keiskei* chloroplast genomes, we identified seven genes that are under positive selection. So, we evaluated a selective analysis of the exons of each of seven protein-coding genes using site-specific models with four comparison models (M0 vs. M3, M1 vs. M2a, M7 vs. M8, M8a vs. M8, likelihood ratio test (LRT) (threshold value $p \leq 0.05$) in EasyCodeML software [26]. Among seven models, M2a is the positive selective model and $p$ ($p_0$, $p_1$ and $p_2$) represents the proportions of negative or purifying, neutral and positive selection. The $\omega_2$ values of seven genes are ranging from 1 to 999 in the M2a model (Supplementary Table S3). Further, we compared these seven genes across the publicly available Nolinoideae chloroplast genome species to understand the selective pressure events. Due to the presence of pseudogenization of the *infA* and *ycf1* gene in some Nolinoideae species, we analyzed the remaining five genes and identified the $\omega_2$ values ranging from 1 to 240.876 (Supplementary Table S4). The variation in $\omega_2$ value might be due to the increase in the number of species analyzed in this study.

To determine which sites are subject to positive selection, Bayes empirical Bayes (BEB) analysis is used to analyze the location of consistent selective sites in the seven protein-coding genes of *S. gardenii* chloroplast genome with its three closely related species using the M7 vs. M8 model. These genes include one NADH-dehydrogenase subunit gene (*ndhF*), one ribosome large subunit gene (*rpl20*), two ribosome small subunit genes (*rps2* and *rps3*) and *ccsA*, *infA* and *ycf1* genes. The analysis of BEB revealed that five sites are under potentially positive selection in the four protein-coding genes (*infA*-3; *ndhF*-1 and *rps3* -1) with posterior probabilities of more than 0.95 and two sites (*rps2*-1 and *ycf1*-1) with greater than 0.99 (Table 2). Furthermore, it could not identify any positively selected sites in the *ccsA* and *rpl20* genes. The 2ΔLnL value of two genes is zero and the *p*-value of LRT is more than 0.05. In contrast, when analyzed with five genes (*ccsA*, *ndhF*, *rpl20*, *rps2* and *rps3*) of all Nolinoideae species, we identified seven sites (*ccsA*-3 and *ndhF*-4) greater than 0.95 and two sites (*ndhF*-1 and *rps2*-1) >0.99. Furthermore, we can able to find out positively selected sites in the *ccsA* gene when analyzed with all the species of the Nolinoideae subfamily. Nevertheless, we could not find any positively selected sites in *rpl20* and *rps3*. In both analyses, *rpl20* does not encode any positively selected sites in their gene, even though it has a higher ω value than *ccsA*, *infA*, *rps2* and *ycf1* genes. To support this analysis, the 2ΔLnL value of *rpl20* is zero and the *p*-value of LRT is greater than 0.05 (Tables 2 and 3). All these highly positive selection genes are involved in the functions of the plant genetic system or photosynthesis process [27–31]. Besides, these seven genes have undergone positive selection, which might be the result of adaptation to their diverse habitats. In the end, highly variable regions and seven protein-coding genes that are identified in their genome to be under positive selection could be used to generate potential markers for phylogenetic studies or candidates for DNA barcoding in future studies.

Simple sequence repeats (SSRs) are extremely powerful molecular marker and play a major role in the population genetics, evolutionary studies, chloroplast genome rearrangement and recombination process [2,18,32–38]. Among the 805 SSRs found in the *S. gardenii* plastome, most of the SSRs are mononucleotide, which occupies 33.29% followed by hexa- (17.39%) and penta- (15.15%) nucleotide SSRs in their genome. We also found many SSRs in IGS regions (64.22%) compared to protein-coding and intron regions in the *S. gardenii* plastome. Because most of the protein-coding genes are highly conserved than intergenic regions of angiosperm chloroplast genomes [3,5,12]. Similarly, the previous studies also revealed that the non-coding region contains more SSRs than the coding regions [18,39,40]. The presence of SSR markers in the *S. Speirantha* could be used to understand the genetic relationships between the closely related species. The complete chloroplast genome sequence-based phylogenetic studies provide essential information regarding the evolutionary relationship among species, genera and families [41–46]. The phylogenetic relationship of the Nolinoideae taxa has not been resolved in earlier studies. The reason for this is that the Nolinoideae subfamily consists of three major tribe clades, namely, Polygonateae, Aspidistreae and Ophiopogoneae, and is weakly supported in the phylogenetic tree [47]. Moreover, in the Polygonateae clade, several *Polygonatum* species were characterized and their phylogenetic relationships have been resolved recently [9,11,14,15]. In contrast, only a few

studies have been carried out in the Aspidistreae tribe clade [10,12,13]. So, in the present study, we constructed a maximum-likelihood phylogenetic tree based on concatenated 75 protein-coding genes of 19 Nolinoideae species and revealed that all the species formed a monophyletic group and *S. gardenii* formed a cluster with *R. carnea*, *R. chinensis* and *C. keiskei*. The previous studies also support this phylogenetic tree where similar results were obtained that showed weakly supported bootstrap value in some nodes of the Nolinoideae phylogeny [9–15,48]. The phylogenetic result also showed that *Speirantha*, *Rohdea*, *Reineckea* and *Convallaria* genus are constantly clustered in the same clade with a high-resolution value, even though high sequence divergence is noted in few protein-coding genes of these chloroplast genomes. These results suggest that *Speirantha* is a sister clade to *Rohdea*, *Reineckea* and *Convallaria* genus. However, the genus *Liriope* made the sister group to the Aspidistreae tribe clade with very weak bootstrap value (26%). Floden and Schilling [47] revealed that the *Maianthemum* is not sister to *Disporopsis*, *Heteropolygonatum* and *Polygonatum* and suggested that *Maianthemum* does not belong to the Polygonateae tribe based on *petA-psbJ* + ITS analyses. In contrast, our results showed that *Maianthemum* is a sister to *Disporopsis*, *Heteropolygonatum* and *Polygonatum* and it belongs to the Polygonateae tribe based on 75 plastid protein-coding genes. To support our results, previous studies also revealed that *Maianthemum* was included in the Polygonateae tribe based on four plastid markers [49,50]. Therefore, more taxa need to be included to understand the phylogenetic position and their relationships with other Nolinoideae subfamily species in future studies.

## 4. Materials and Methods

### 4.1. DNA Extraction and Sequencing

A high-quality *Speirantha gardenii* DNA sample was obtained from the DNA bank of the Royal Botanic Gardens, Kew, London, England (http://data.kew.org/dnabank/DnaBankForm.html). Whole-genome sequencing was performed using Illumina HiSeq2500 (Phyzen Ltd., South Korea) and a paired-end (PE) library of 2 × 150 bp and an insert size of ~550 bp and obtained 8,363,058,594 raw reads. Read quality was analyzed with FastQC [51] and low-quality reads were removed with Trimmomatic 0.39 [52]. The clean reads were filtered with GetOrganelle pipe-line (https://github.com/Kinggerm/GetOrganelle) to get plastid-like reads, then the filtered reads were assembled by de nova approach using SPAdes version 3.12.0 [53]. The obtained contigs (>500 bp) were mapped with *Nicotiana tabacum* plastid genes (NC_001879) and three putative plastid-like contigs were identified and scaffolded using Geneious Prime (Biomatters, New Zealand). complete chloroplast genome sequence and gene annotation were submitted to GenBank and assigned the accession number MT797212.

### 4.2. The Chloroplast Genome Annotation of the S. gardenii

Dual Organeller GenoMe Annotator (DOGMA) program was employed to annotate the *S. gardenii* chloroplast genome [54]. The initial annotation, putative starts, stops, and intron positions were adjusted by comparing them with closely related *Convallaria keiskei* homologous genes. Transfer RNA genes were verified using tRNAscan-SE version1.21 with default settings [55]. The online program OGDRAW was used to draw a circular map of the *S. gardenii* chloroplast genome [56].

### 4.3. Comparative Chloroplast Genome Analysis of the S. gardenii

The mVISTA program in Shuffle-LAGAN mode was used to compare the *S. gardenii* chloroplast genome with closely related three other chloroplast genomes namely *Reineckea carnea*, *Rohdea chinensis* and *C. keiskei* using *S. gardenii* annotation as a reference [57]. The boundaries between IR and SC regions of these species were also compared and analyzed.

*4.4. Characterization of Substitution Rates*

To evaluate synonymous ($K_S$) and nonsynonymous ($K_A$) substitution rates, the *S. gardenii* chloroplast genome was compared with the three other chloroplast genome sequences of *R. carnea*, *R. chinensis* and *C. keiskei*. The similar individual functional protein-coding gene exons of these genomes were extracted and aligned separately using Geneious Prime (Biomatters, New Zealand). The aligned sequences were translated into protein sequences and substitution rates were analyzed using DnaSP [58].

*4.5. Positive Selection Analysis*

To detect the nonsynonymous vs. synonymous ($\omega$) ratio of seven protein-coding genes (*ccsA*, *infA*, *ndhF*, *rpl20*, *rps2*, *rps3* and *ycf1*) under selection in four species *S. gardenii*, *R. carnea*, *R. chinensis* and *C. keiskei* and all Nolinoideae species separately, the sequences of each gene were aligned using the MAFFT program [59] and the maximum likelihood phylogenetic tree was constructed using RAxML v. 7.2.6 [60]. The nested site-specific model was conducted to calculate nonsynonymous ($K_A$) and synonymous substitution ($K_S$) ratio using EasyCodeML [26]. The seven codon substitution models described as M0, M1a, M2a, M3, M7, M8 and M8a were examined. Two likelihood ratio tests were conducted to identify the positively selected sites: M0 (one-ratio) vs. M3 (discrete), M1a (neutral) vs. M2a (positive selection) and M7 ($\beta$) vs. M8 ($\beta$ and $\omega > 1$) and M8a (($\beta$ and $\omega = 1$) vs. M8, which were compared using a nested site-specific model [26]. The likelihood ratio test (LRT) of the above comparison was carried out respectively to evaluate the selection strength and the *p*-values of Chi-square ($x^2$) lesser than 0.05 were considered as significant. If the LRT *p*-values were significant (<0.05), Bayes empirical Bayes (BEB) method was implemented to identify codons under positive selection. The BEB values higher than 0.95 and 0.99 indicate the sites potentially under positive selection and highly positive selection, which is indicated by asterisks and double asterisks, respectively.

*4.6. Analysis of Repeat Sequences and Single Sequence Repeats (SSR)*

REPuter software was applied to detect the presence of repeat sequences, including forward, reverse, palindromic and complementary repeats in the chloroplast genome of *S. gardenii* [61]. The following parameters were used to identify repeats in REPuter: (1) Hamming distance 3, (2) minimum sequence identity of 90%, (3) and a repeat size of more than 30 bp. Phobos software v1.0.6 was employed to discover SSRs of chloroplast genome; parameters for the match, mismatch, gap and N positions were set at 1, −5, −5 and 0, respectively [62].

*4.7. Phylogenetic Tree Analysis*

A phylogenetic tree was constructed using 75 protein-coding genes of 19 Nolinoideae chloroplast genomes and *Asparagus* used as the outgroup. The 18 completed chloroplast genome sequences were downloaded from the National Center for Biotechnology Information (NCBI) Organelle Genome Resource database (Supplementary Table S5). The aligned protein-coding gene sequences were saved in PHYLogeny Inference Package (PHYLIP) format using Clustal X v2.1 [63] and phylogenetic analysis was constructed based on maximum likelihood (ML) analysis using the GTRI model by RAxML v. 7.2.6 with 1000 bootstrap replications [60].

## 5. Conclusions

The present study describes the complete chloroplast genome sequence of *Speirantha gardenii* and is the first such report for a species in *Speirantha*. The *S. gardenii* chloroplast genome (156,869 bp) is fully characterized and compared with its closely related species of Nolinoideae subfamily. Overall, the gene contents and gene arrangements are similar and highly conserved in the species of Nolinoideae subfamily. Furthermore, high sequence variation in the protein-coding and intergenic regions, repeat sequences analysis, nucleotide substitution patterns and amino acid sites under potentially positive

selection in seven protein-coding genes in the chloroplast genomes of species of the Nolinoideae subfamily may be useful for developing a lineage-specific marker for genetic diversity and gene evolution studies. Besides, phylogenomic studies showed that the genera *Speirantha*, *Rohdea*, *Reineckea* and *Convallaria* are constantly clustered in the same clade suggest that these taxa are close genetic relationships to each other and highly conserved in the Nolinoideae subfamily.

**Supplementary Materials:** Supplementary Materials can be found at http://www.mdpi.com/2073-4395/10/9/1405/s1. Table S1: List of genes present in the chloroplast genome of *Speirantha gardenii*. Supplementary, Table S2: Comparison of general features of Nolinoideae chloroplast genomes, Table S3, Comparison of site models, positive selective amino acid loci and estimation of parameters for seven protein-coding genes in the four Nolinoideae species, Table S4: Comparison of site models, positive selective amino acid loci and estimation of parameters for five protein-coding genes in the Nolinoideae species, Table S5. List of chloroplast genomes used for phylogenetic analysis.

**Author Contributions:** S.P. and G.R. conceived and designed the experiments. G.R. performed the experiments, analyzed the data, and prepared a draft of the manuscript and figures. S.P. and G.R. modified the manuscript. All authors have read and agreed to the published version of the manuscript.

**Funding:** The National Research Foundation of Korea (NRF) (2019R1F1A1062102) project, Ministry of Education, the Republic of Korea, awarded to Gurusamy Raman, Department of Life Sciences, Yeungnam University.

**Acknowledgments:** This study was supported by the National Research Foundation of Korea (NRF) (2019R1F1A1062102) project, Ministry of Education, Republic of Korea, awarded to Gurusamy Raman, Department of Life Sciences, Yeungnam University.

**Conflicts of Interest:** The authors declare that they have no competing interests.

## Abbreviations

| | |
|---|---|
| LSC | Large single-copy |
| SSC | Small single-copy |
| IRs | Inverted repeats |
| CNVs | Copy number variations |
| DOGMA | Dual Organeller GenoMe Annotator |
| tRNA | Transfer RNA |
| rRNA | Ribosomal RNA |
| $K_S$ | Synonymous substitution |
| $K_A$ | Non-synonymous substitution |
| $\omega$ | Nonsynonymous vs. synonymous ratio |
| SSR | Simple sequence repeats |
| LRT | Likelihood ratio test |

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
