# Peer review of "The Complete Chloroplast Genome Sequence of the Speirantha gardenii: Comparative and Adaptive Evolutionary Analysis"

_agronomy, doi:10.3390/agronomy10091405_

Round 1

Reviewer 1 Report

This manuscript reported the complete chloroplast genome sequence of Speirantha gardenia. The authors also compared the characterized S. gardenii cp genome to its closely related Nolinoideae species.  I only have some minor concerns.

  1. The resolution of figure 1 is not good. When I tried to enlarge the figure, it’s difficult to recognize the letters.
  2. It would be nice to provide more details in the materials and methods such as how to conduct sequence assembly.
  3. In the title, the scientific name “Speirantha gardenia” should be in the italic form. And I am not sure whether the “C” in the word “Comparative” should be in capital letter or not.

Author Response

Point 1: The resolution of figure 1 is not good. When I tried to enlarge the figure, it’s difficult to recognize the letters.

Response 1: Yes. We have provided a high-resolution figure (300 dpi) as a separate file in the revised manuscript.

Point 2: It would be nice to provide more details in the materials and methods such as how to conduct sequence assembly.

Response 2: We have incorporated the sequence assembly strategy in the revised manuscript (L. 340-342).

Point 3: In the title, the scientific name “Speirantha gardenia” should be in the italic form. And I am not sure whether the “C” in the word “Comparative” should be in capital letter or not.

Response 3: Yes. We have changed “Speirantha gardenia” into “Speirantha gardenia” and the word “Comparative” into the sentence form letter in the revised manuscript (L. 3)

Reviewer 2 Report

Dear authors,

Introduction is a little too short. Suggest to provide more background for this topic.

Results are clear.

Discussion: Generally there are no big issues, but if you can add more citations to make more comparison for the last paragraph, that will be better.

Methodology has no problems.

Citations has no problem.

  1. Please edit some English grammar errors.
  2. Line 11: “So” changes into Thus.
  3. Line 363: “species” changes into taxa is better.
  4. Line 289: Suggest to add more citations to make the comparison.

Author Response

Point 1: Introduction is a little too short. Suggest to provide more background for this topic.

Response 1: Yes. We have incorporated additional background information in the revised manuscript L. 32-36; 41-6)

Point 2: Generally there are no big issues, but if you can add more citations to make more comparison for the last paragraph, that will be better.

Response 2: Yes. We have discussed more detail and included citations in the discussion part of the revised manuscript (L. 300-304; 306-312; 315-317; 321-330)

Point 3: Line 11: “So” changes into Thus.

Response 3: Yes. We have revised the sentence in the revised manuscript (L.11)

Point 4: Line 363: “species” changes into taxa is better

Response 4: Yes. “species” has been changed into “taxa” in the revised manuscript (L. 405).

Point 5: Line 289: Suggest to add more citations to make the comparison.

Response 5: Yes. The phylogenetic tree has been compared with previous studies and incorporated more citations in the revised manuscript (L. 306-330).

Reviewer 3 Report

In this manuscript, Raman and Park report their work sequencing and analyzing the plastid genome of Speirantha gardenii, a previously unsequenced genus of monocot.  They find, as expected, that the plastid genome sequence of S. gardenii is very similar to that of its close relatives in the Nolinoideae, and use comparative approaches with several closely-related genera (Reineckea, Rohdea, and Covallaria) to uncover similarities and differences in plastid genome evolution in this clade, as well as sites of positive, neutral, or negative selection.  The authors are appropriately modest in this report, providing clear figures and reasonably clear explanation of their data, and offering a new resource for phylogeneticists interested in resolving the relationships of this branch of monocots (which has been somewhat contested in the past ~10-15 years).

My only major concern is that the manuscript requires some degree of editing for English language.  I can almost always follow the paper, but it can be challenging in some places to read, so a quick edit from a colleague with the necessary language skills would really benefit the paper and make it more accessible to a wide audience.

A few more points:

(1) Nolinoideae is not a "family", but a "subfamily" of the Asparagaceae.  This should be corrected throughout.

(2) "cp genome" adds another acronym that isn't needed.  I recommend simply saying "chloroplast genome" (or, more correctly but admittedly less common in the literature, "plastid genome") throughout.

(3) Since this plastid genome comes from a botanical garden specimen, the manuscript should clarify that this genome may not represent the diversity of sequences in the species.  While this could be used as a reference for future comparison, without some attempt at population genetics to determine the variability of these sequences, a caveat must be added.

(4) How many plants were sampled?  The methods say "a high quality S. gardenii DNA sample was obtained": what tissue(s) were used, from how many individual plants, and how was the DNA extracted and purified?

(5) In the conclusions, the authors say "...suggest that these species are highly conserved."  Species cannot be "conserved".  They might be "closely related."  This is, in fact, one example of how English language could create confusion--the point is not that the genome is conserved (of course plastid genomes are conserved), but that the genome shows very little variation from the related species.

(6) If it is possible to obtain additional samples from individual plants at other locations and resequence some sections of the genome--especially, say, those regions that are proposed to be under selection--that would add significantly to the potential impact and use of this report in the future.  I do understand that obtaining those samples might be difficult, but resequencing with Sanger approaches would be pretty straightforward.

Author Response

Point 1: Nolinoideae is not a "family", but a "subfamily" of the Asparagaceae.  This should be corrected throughout.

Response 1: Yes. It has been revised throughout in the revised manuscript.

Point 2: "cp genome" adds another acronym that isn't needed.  I recommend simply saying "chloroplast genome" (or, more correctly but admittedly less common in the literature, "plastid genome") throughout.

Response 2: Yes. It has been changed into “chloroplast” in the whole revised manuscript.

Point 3: Since this plastid genome comes from a botanical garden specimen, the manuscript should clarify that this genome may not represent the diversity of sequences in the species.  While this could be used as a reference for future comparison, without some attempt at population genetics to determine the variability of these sequences, a caveat must be added.

Response 3: Yes. The Kew Gardens mentioned that they have used only one fresh young leave sample for DNA isolation and this DNA sample does not contain a diversity of sequences. Since the endangered, rare availability, the genus Speirantha contain only one species and no other studies for this species, we were interested and approached Kew Gardens to obtain plants. But due to the Kew Gardens policy, they have not provided plants. So, we obtained DNA for this study and this is the first report. So, this sequencing data can be used as reference data for population studies and future work will be carried with this genus. Also, we approached Kew Gardens to obtain information for these queries and responded. I am herewith enclosing their response for your reference.

Point 4: How many plants were sampled?  The methods say "a high quality S. gardenii DNA sample was obtained": what tissue(s) were used, from how many individual plants, and how was the DNA extracted and purified?

Response 4: Kew Gardens isolated DNA young fresh leaf tissue of one individual plant. The DNA extraction was performed by using a modified 2x CTAB method, followed by CsCl/EtBr gradient and dialysis. This method has yielded the best and high-quality DNA from plant tissue. Since we obtained a DNA sample from Kew Gardens, we cannot incorporate sample collection and DNA extraction methods in the manuscript.

Point 5: In the conclusions, the authors say "...suggest that these species are highly conserved."  Species cannot be "conserved".  They might be "closely related."  This is, in fact, one example of how English language could create confusion--the point is not that the genome is conserved (of course plastid genomes are conserved), but that the genome shows very little variation from the related species.

Response 5: Yes. The sentence has been revised in the revised manuscript (L. 404-406).

Point 6: If it is possible to obtain additional samples from individual plants at other locations and resequence some sections of the genome--especially, say, those regions that are proposed to be under selection--that would add significantly to the potential impact and use of this report in the future.  I do understand that obtaining those samples might be difficult, but resequencing with Sanger approaches would be pretty straightforward

Response 6: We have requested a DNA sample from Kew Gardens for the present study. But we have received it after four months. Due to the COVID-19 pandemic situation, the Kew Garden has been closed and the limited availability of staff, it’s difficult to get a DNA sample at this moment. Also, this plant is endemic to China, it is difficult to obtain a plant sample now.
